# SUB-POLICY ADAPTATION FOR HIERARCHICAL REINFORCEMENT LEARNING

**Alexander C. Li**[*]**, Carlos Florensa**[*]**, Ignasi Clavera, Pieter Abbeel**
University of California, Berkeley
`{alexli1, florensa, iclavera, pabbeel}@berkeley.edu`

## ABSTRACT

Hierarchical reinforcement learning is a promising approach to tackle long-horizon decision-making problems with sparse rewards. Unfortunately, most methods still decouple the lower-level skill acquisition process and the training of a higher level that controls the skills in a new task. Leaving the skills fixed can lead to significant sub-optimality in the transfer setting. In this work, we propose a novel algorithm to discover a set of skills and continuously adapt them along with the higher level even when training on a new task. Our main contributions are two-fold. First, we derive a new hierarchical policy gradient with an unbiased latent-dependent baseline, and we introduce Hierarchical Proximal Policy Optimization (HiPPO), an on-policy method to efficiently train all levels of the hierarchy jointly. Second, we propose a method of training time-abstractions that improves the robustness of the obtained skills to environment changes. Code and videos are available. [1].

## 1 INTRODUCTION

Reinforcement learning (RL) has made great progress in a variety of domains, from playing games such as Pong and Go (Mnih et al., 2015; Silver et al., 2017) to automating robotic locomotion (Schulman et al., 2015; Heess et al., 2017), dexterous manipulation (Florensa et al., 2017b; OpenAI et al., 2018), and perception (Nair et al., 2018; Florensa et al., 2018). Yet, most work in RL is still learning from scratch when faced with a new problem. This is particularly inefficient when tackling multiple related tasks that are hard to solve due to sparse rewards or long horizons.

A promising technique to overcome this limitation is hierarchical reinforcement learning (HRL) (Sutton et al., 1999). In this paradigm, policies have several modules of abstraction, allowing to reuse subsets of the modules. The most common case consists of temporal hierarchies (Precup, 2000; Dayan & Hinton, 1993), where a higher-level policy (manager) takes actions at a lower frequency, and its actions condition the behavior of some lower level skills or sub-policies. When transferring knowledge to a new task, most prior works fix the skills and train a new manager on top. Despite having a clear benefit in kick-starting the learning in the new task, having fixed skills can considerably cap the final performance on the new task (Florensa et al., 2017a). Little work has been done on adapting pre-trained sub-policies to be optimal for a new task.

In this paper, we develop a new framework for simultaneously adapting all levels of temporal hierarchies. First, we derive an efficient approximated hierarchical policy gradient. The key insight is that, despite the decisions of the manager being unobserved latent variables from the point of view of the Markovian environment, from the perspective of the sub-policies they can be considered as part of the observation. We show that this provides a decoupling of the manager and sub-policy gradients, which greatly simplifies the computation in a principled way. It also theoretically justifies a technique used in other prior works (Frans et al., 2018). Second, we introduce a sub-policy specific baseline for our hierarchical policy gradient. We prove that this baseline is unbiased, and our experiments reveal faster convergence, suggesting efficient gradient variance reduction. Then, we introduce a more stable way of using this gradient, Hierarchical Proximal Policy Optimization (HiPPO). This method helps us take more conservative steps in our policy space (Schulman et al., 2017), critical in hierarchies

---

[1] `sites.google.com/view/hippo-rl`

[*] Equal Contribution

because of the interdependence of each layer. Results show that HiPPO is highly efficient both when learning from scratch, i.e. adapting randomly initialized skills, and when adapting pretrained skills on a new task. Finally, we evaluate the benefit of randomizing the time-commitment of the sub-policies, and show it helps both in terms of final performance and zero-shot adaptation on similar tasks.

## 2 PRELIMINARIES

We define a discrete-time finite-horizon discounted Markov decision process (MDP) by a tuple $M = (\mathcal{S}, \mathcal{A}, \mathcal{P}, r, \rho_0, \gamma, H)$, where $\mathcal{S}$ is a state set, $\mathcal{A}$ is an action set, $\mathcal{P} : \mathcal{S} \times \mathcal{A} \times \mathcal{S} \to \mathbb{R}_+$ is the transition probability distribution, $\gamma \in [0, 1]$ is a discount factor, and $H$ the horizon. Our objective is to find a stochastic policy $\pi_\theta$ that maximizes the expected discounted return within the MDP, $\eta(\pi_\theta) = \mathbb{E}_\tau [\sum_{t=0}^{H} \gamma^t r(s_t, a_t)]$. We use $\tau = (s_0, a_0, ...,)$ to denote the entire state-action trajectory, where $s_0 \sim \rho_0(s_0)$, $a_t \sim \pi_\theta(a_t|s_t)$, and $s_{t+1} \sim \mathcal{P}(s_{t+1}|s_t, a_t)$.

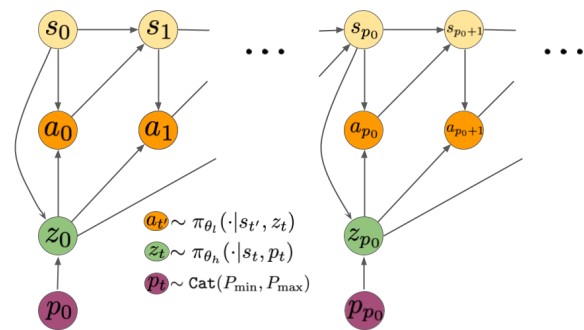

Figure 1: Temporal hierarchy studied in this paper. A latent code $z_t$ is sampled from the manager policy $\pi_{\theta_h}(z_t|s_t)$ every $p$ time-steps, using the current observation $s_{kp}$. The actions $a_t$ are sampled from the sub-policy $\pi_{\theta_l}(a_t|s_t, z_{kp})$ conditioned on the same latent code from $t = kp$ to $(k+1)p - 1$

In this work, we propose a method to learn a hierarchical policy and efficiently adapt all the levels in the hierarchy to perform a new task. We study hierarchical policies composed of a higher level, or manager $\pi_{\theta_h}(z_t|s_t)$, and a lower level, or sub-policy $\pi_{\theta_l}(a_{t'}|z_t, s_{t'})$. The higher level does not take actions in the environment directly, but rather outputs a command, or latent variable $z_t \in \mathcal{Z}$, that conditions the behavior of the lower level. We focus on the common case where $\mathcal{Z} = \mathbb{Z}_n$ making the manager choose among $n$ sub-policies, or skills, to execute. The manager typically operates at a lower frequency than the sub-policies, only observing the environment every $p$ time-steps. When the manager receives a new observation, it decides which low level policy to commit to for $p$ environment steps by the means of a latent code $z$. Figure 1 depicts this framework where the high level frequency $p$ is a random variable, which is one of the contribution of this paper as described in Section 4.4. Note that the class of hierarchical policies we work with is more restrictive than others like the options framework, where the time-commitment is also decided by the policy. Nevertheless, we show that this loss in policy expressivity acts as a regularizer and does not prevent our algorithm from surpassing other state-of-the art methods.

## 3 RELATED WORK

There has been growing interest in HRL for the past few decades (Sutton et al., 1999; Precup, 2000), but only recently has it been applied to high-dimensional continuous domains as we do in this work (Kulkarni et al., 2016; Daniel et al., 2016). To obtain the lower level policies, or skills, most methods exploit some additional assumptions, like access to demonstrations (Le et al., 2018; Merel et al., 2019; Ranchod et al., 2015; Sharma et al., 2018), policy sketches (Andreas et al., 2017), or task decomposition into sub-tasks (Ghavamzadeh & Mahadevan, 2003; Sohn et al., 2018). Other methods use a different reward for the lower level, often constraining it to be a "goal reacher" policy, where the signal from the higher level is the goal to reach (Nachum et al., 2018; Levy et al., 2019; Vezhnevets et al., 2017). These methods are very promising for state-reaching tasks, but might require access to goal-reaching reward systems not defined in the original MDP, and are more limited when training on tasks beyond state-reaching. Our method does not require any additional supervision, and the obtained skills are not constrained to be goal-reaching.

When transferring skills to a new environment, most HRL methods keep them fixed and simply train a new higher-level on top (Hausman et al., 2018; Heess et al., 2016). Other work allows for building on previous skills by constantly supplementing the set of skills with new ones (Shu et al., 2018), but they require a hand-defined curriculum of tasks, and the previous skills are never fine-tuned.

Our algorithm allows for seamless adaptation of the skills, showing no trade-off between leveraging the power of the hierarchy and the final performance in a new task. Other methods use invertible functions as skills (Haarnoja et al., 2018), and therefore a fixed skill can be fully overwritten when a new layer of hierarchy is added on top. This kind of "fine-tuning" is promising, although similar to other works (Peng et al., 2019), they do not apply it to temporally extended skills as we do here.

One of the most general frameworks to define temporally extended hierarchies is the options framework (Sutton et al., 1999), and it has recently been applied to continuous state spaces (Bacon et al., 2017). One of the most delicate parts of this formulation is the termination policy, and it requires several regularizers to avoid skill collapse (Harb et al., 2017; Vezhnevets et al., 2016). This modification of the objective may be difficult to tune and affects the final performance. Instead of adding such penalties, we propose to have skills of a random length, not controlled by the agent during training of the skills. The benefit is two-fold: no termination policy to train, and more stable skills that transfer better. Furthermore, these works only used discrete action MDPs. We lift this assumption, and show good performance of our algorithm in complex locomotion tasks. There are other algorithms recently proposed that go in the same direction, but we found them more complex, less principled (their per-action marginalization cannot capture well the temporal correlation within each option), and without available code or evidence of outperforming non-hierarchical methods (Smith et al., 2018).

The closest work to ours in terms of final algorithm structure is the one proposed by Frans et al. (2018). Their method can be included in our framework, and hence benefits from our new theoretical insights. We introduce a modification that is shown to be highly beneficial: the random time-commitment mentioned above, and find that our method can learn in difficult environments without their complicated training scheme.

## 4 EFFICIENT HIERARCHICAL POLICY GRADIENTS

When using a hierarchical policy, the intermediate decision taken by the higher level is not directly applied in the environment. Therefore, technically it should not be incorporated into the trajectory description as an observed variable, like the actions. This makes the policy gradient considerably harder to compute. In this section we first prove that, under mild assumptions, the hierarchical policy gradient can be accurately approximated without needing to marginalize over this latent variable. Then, we derive an unbiased baseline for the policy gradient that can reduce the variance of its estimate. Finally, with these findings, we present our method, Hierarchical Proximal Policy Optimization (HiPPO), an on-policy algorithm for hierarchical policies, allowing learning at all levels of the policy jointly and preventing sub-policy collapse.

### 4.1 APPROXIMATE HIERARCHICAL POLICY GRADIENT

Policy gradient algorithms are based on the likelihood ratio trick (Williams, 1992) to estimate the gradient of returns with respect to the policy parameters as

$$\nabla_\theta \eta(\pi_\theta) = \mathbb{E}_\tau \left[ \nabla_\theta \log P(\tau) R(\tau) \right] \approx \frac{1}{N} \sum_{i=1}^{n} \nabla_\theta \log P(\tau_i) R(\tau_i) \tag{1}$$

$$= \frac{1}{N} \sum_{i=1}^{n} \frac{1}{H} \sum_{t=1}^{H} \nabla_\theta \log \pi_\theta(a_t|s_t) R(\tau_i) \tag{2}$$

In a temporal hierarchy, a hierarchical policy with a manager $\pi_{\theta_h}(z_t|s_t)$ selects every $p$ time-steps one of $n$ sub-policies to execute. These sub-policies, indexed by $z \in \mathbb{Z}_n$, can be represented as a single conditional probability distribution over actions $\pi_{\theta_l}(a_t|z_t, s_t)$. This allows us to not only use a given set of sub-policies, but also leverage skills learned with Stochastic Neural Networks (SNNs) (Florensa et al., 2017a). Under this framework, the probability of a trajectory $\tau = (s_0, a_0, s_1, \ldots, s_H)$ can be written as

$$P(\tau) = \left( \prod_{k=0}^{H/p} \left[ \sum_{j=1}^{n} \pi_{\theta_h}(z_j|s_{kp}) \prod_{t=kp}^{(k+1)p-1} \pi_{\theta_l}(a_t|s_t, z_j) \right] \right) \left[ P(s_0) \prod_{t=1}^{H} P(s_{t+1}|s_t, a_t) \right]. \tag{3}$$

The mixture action distribution, which presents itself as an additional summation over skills, prevents additive factorization when taking the logarithm, as from Eq. 1 to 2. This can yield numerical

instabilities due to the product of the $p$ sub-policy probabilities. For instance, in the case where all the skills are distinguishable all the sub-policies' probabilities but one will have small values, resulting in an exponentially small value. In the following Lemma, we derive an approximation of the policy gradient, whose error tends to zero as the skills become more diverse, and draw insights on the interplay of the manager actions.

**Lemma 1.** *If the skills are sufficiently differentiated, then the latent variable can be treated as part of the observation to compute the gradient of the trajectory probability. Let $\pi_{\theta_h}(z|s)$ and $\pi_{\theta_l}(a|s,z)$ be Lipschitz functions w.r.t. their parameters, and assume that $0 < \pi_{\theta_l}(a|s,z_j) < \epsilon \; \forall j \neq kp$, then*

$$\nabla_\theta \log P(\tau) = \sum_{k=0}^{H/p} \nabla_\theta \log \pi_{\theta_h}(z_{kp}|s_{kp}) + \sum_{t=0}^{H} \nabla_\theta \log \pi_{\theta_l}(a_t|s_t, z_{kp}) + \mathcal{O}(nH\epsilon^{p-1}) \quad (4)$$

*Proof.* See Appendix. $\qquad\qquad\square$

Our assumption can be seen as having diverse skills. Namely, for each action there is just one sub-policy that gives it high probability. In this case, the latent variable can be treated as part of the observation to compute the gradient of the trajectory probability. Many algorithms to extract lower-level skills are based on promoting diversity among the skills (Florensa et al., 2017a; Eysenbach et al., 2019), therefore usually satisfying our assumption. We further analyze how well this assumption holds in our experiments section and Table 2.

## 4.2 Unbiased Sub-Policy Baseline

The policy gradient estimate obtained when applying the log-likelihood ratio trick as derived above is known to have large variance. A very common approach to mitigate this issue without biasing the estimate is to subtract a baseline from the returns (Peters & Schaal, 2008). It is well known that such baselines can be made state-dependent without incurring any bias. However, it is still unclear how to formulate a baseline for all the levels in a hierarchical policy, since an action dependent baseline does introduce bias in the gradient (Tucker et al., 2018). It has been recently proposed to use latent-conditioned baselines (Weber et al., 2019). Here we go further and prove that, under the assumptions of Lemma 1, we can formulate an unbiased latent dependent baseline for the approximate gradient (Eq. 5).

**Lemma 2.** *For any functions $b_h : \mathcal{S} \to \mathbb{R}$ and $b_l : \mathcal{S} \times \mathcal{Z} \to \mathbb{R}$ we have:*

$$\mathbb{E}_\tau[\sum_{k=0}^{H/p} \nabla_\theta \log P(z_{kp}|s_{kp}) b_h(s_{kp})] = 0 \quad and \quad \mathbb{E}_\tau[\sum_{t=0}^{H} \nabla_\theta \log \pi_{\theta_l}(a_t|s_t, z_{kp}) b_l(s_t, z_{kp})] = 0$$

*Proof.* See Appendix. $\qquad\qquad\square$

Now we apply Lemma 1 and Lemma 2 to Eq. 1. By using the corresponding value functions as the function baseline, the return can be replaced by the Advantage function $A(s_{kp}, z_{kp})$ (see details in Schulman et al. (2016)), and we obtain the following approximate policy gradient expression:

$$\hat{g} = \mathbb{E}_\tau\Big[(\sum_{k=0}^{H/p} \nabla_\theta \log \pi_{\theta_h}(z_{kp}|s_{kp}) A(s_{kp}, z_{kp})) + (\sum_{t=0}^{H} \nabla_\theta \log \pi_{\theta_l}(a_t|s_t, z_{kp}) A(s_t, a_t, z_{kp}))\Big]$$

This hierarchical policy gradient estimate can have lower variance than without baselines, but using it for policy optimization through stochastic gradient descent still yields an unstable algorithm. In the next section, we further improve the stability and sample efficiency of the policy optimization by incorporating techniques from Proximal Policy Optimization (Schulman et al., 2017).

## 4.3 Hierarchical Proximal Policy Optimization

Using an appropriate step size in policy space is critical for stable policy learning. Modifying the policy parameters in some directions may have a minimal impact on the distribution over actions, whereas small changes in other directions might change its behavior drastically and hurt training

---

**Algorithm 1** HiPPO Rollout

1: **Input:** skills $\pi_{\theta_l}(a|s,z)$, manager $\pi_{\theta_h}(z|s)$, time-commitment bounds $P_{\min}$ and $P_{\max}$, horizon $H$
2: Reset environment: $s_0 \sim \rho_0$, $t = 0$.
3: **while** $t < H$ **do**
4:     Sample time-commitment $p \sim \mathtt{Cat}([P_{\min}, P_{\max}])$
5:     Sample skill $z_t \sim \pi_{\theta_h}(\cdot|s_t)$
6:     **for** $t' = t \ldots (t+p)$ **do**
7:         Sample action $a_{t'} \sim \pi_{\theta_l}(\cdot|s_{t'}, z_t)$
8:         Observe new state $s_{t'+1}$ and reward $r_{t'}$
9:     **end for**
10:    $t \leftarrow t + p$
11: **end while**
12: **Output:** $(s_0, z_0, a_0, s_1, a_1, \ldots, s_H, z_H, a_H, s_{H+1})$

**Algorithm 2** HiPPO

1: **Input:** skills $\pi_{\theta_l}(a|s,z)$, manager $\pi_{\theta_h}(z|s)$, horizon $H$, learning rate $\alpha$
2: **while** not done **do**
3:     **for** actor = 1, 2, ..., N **do**
4:         Obtain trajectory with HiPPO Rollout
5:         Estimate advantages $\hat{A}(a_{t'}, s_{t'}, z_t)$ and $\hat{A}(z_t, s_t)$
6:     **end for**
7:     $\theta \leftarrow \theta + \alpha \nabla_\theta L_{HiPPO}^{CLIP}(\theta)$
8: **end while**

---

efficiency (Kakade, 2002). Trust region policy optimization (TRPO) uses a constraint on the KL-divergence between the old policy and the new policy to prevent this issue (Schulman et al., 2015). Unfortunately, hierarchical policies are generally represented by complex distributions without closed form expressions for the KL-divergence. Therefore, to improve the stability of our hierarchical policy gradient we turn towards Proximal Policy Optimization (PPO) (Schulman et al., 2017). PPO is a more flexible and compute-efficient algorithm. In a nutshell, it replaces the KL-divergence constraint with a cost function that achieves the same trust region benefits, but only requires the computation of the likelihood. Letting $w_t(\theta) = \frac{\pi_\theta(a_t|s_t)}{\pi_{\theta_{old}}(a_t|s_t)}$, the PPO objective is:

$$L^{CLIP}(\theta) = \mathbb{E}_t \min \left\{ w_t(\theta) A_t, \ \mathtt{clip}(w_t(\theta), 1 - \epsilon, 1 + \epsilon) A_t \right\}$$

We can adapt our approximated hierarchical policy gradient with the same approach by letting $w_{h,kp}(\theta) = \frac{\pi_{\theta_h}(z_{kp}|s_{kp})}{\pi_{\theta_{h,old}}(z_{kp}|s_{kp})}$ and $w_{l,t}(\theta) = \frac{\pi_{\theta_l}(a_t|s_t,z_{kp})}{\pi_{\theta_{l,old}}(a_t|s_t,z_{kp})}$, and using the super-index $\mathtt{clip}$ to denote the clipped objective version, we obtain the new surrogate objective:

$$
\begin{aligned}
L_{HiPPO}^{CLIP}(\theta) = \mathbb{E}_\tau \Big[ &\sum_{k=0}^{H/p} \min \left\{ w_{h,kp}(\theta) A(s_{kp}, z_{kp}), w_{h,kp}^{\mathtt{clip}}(\theta) A(s_{kp}, z_{kp}) \right\} \\
&+ \sum_{t=0}^{H} \min \left\{ w_{l,t}(\theta) A(s_t, a_t, z_{kp}), w_{l,t}^{\mathtt{clip}}(\theta) A(s_t, a_t, z_{kp}) \right\} \Big]
\end{aligned}
$$

We call this algorithm Hierarchical Proximal Policy Optimization (HiPPO). Next, we introduce a critical additions: a switching of the time-commitment between skills.

## 4.4 Varying Time-commitment

Most hierarchical methods either consider a fixed time-commitment to the lower level skills (Florensa et al., 2017a; Frans et al., 2018), or implement the complex options framework (Precup, 2000; Bacon et al., 2017). In this work we propose an in-between, where the time-commitment to the skills is a random variable sampled from a fixed distribution $\mathtt{Categorical}(T_{\min}, T_{\max})$ just before the manager takes a decision. This modification does not hinder final performance, and we show it improves zero-shot adaptation to a new task. This approach to sampling rollouts is detailed in Algorithm 1. The full algorithm is detailed in Algorithm 2.

## 5 Experiments

We designed our experiments to answer the following questions: 1) How does HiPPO compare against a flat policy when learning from scratch? 2) Does it lead to policies more robust to environment changes? 3) How well does it adapt already learned skills? and 4) Does our skill diversity assumption hold in practice?

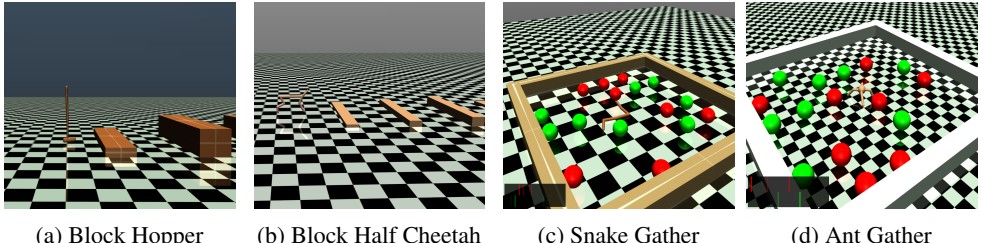

| (a) Block Hopper | (b) Block Half Cheetah | (c) Snake Gather | (d) Ant Gather |

Figure 2: Environments used to evaluate the performance of our method. Every episode has a different configuration: wall heights for (a)-(b), ball positions for (c)-(d)

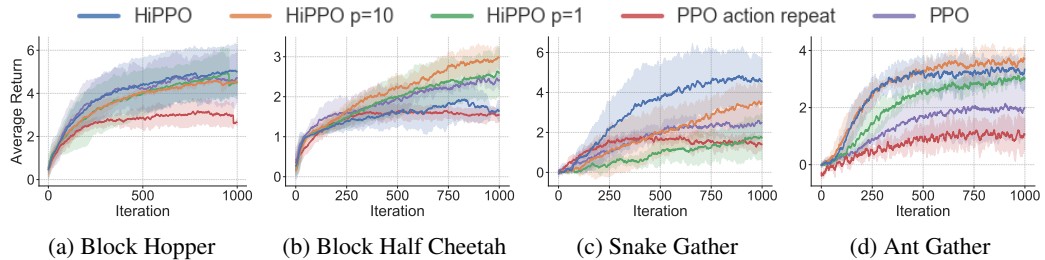

| (a) Block Hopper | (b) Block Half Cheetah | (c) Snake Gather | (d) Ant Gather |

Figure 3: Analysis of different time-commitment strategies on learning from scratch.

## 5.1 TASKS

We evaluate our approach on a variety of robotic locomotion and navigation tasks. The Block environments, depicted in Fig. 2a-2b, have walls of random heights at regular intervals, and the objective is to learn a gait for the Hopper and Half-Cheetah robots to jump over them. The agents observe the height of the wall ahead and their proprioceptive information (joint positions and velocities), receiving a reward of +1 for each wall cleared. The Gather environments, described by Duan et al. (2016), require agents to collect apples (green balls, +1 reward) while avoiding bombs (red balls, -1 reward). The only available perception beyond proprioception is through a LIDAR-type sensor indicating at what distance are the objects in different directions, and their type, as depicted in the bottom left corner of Fig. 2c-2d. This is challenging hierarchical task with sparse rewards that requires simultaneously learning perception, locomotion, and higher-level planning capabilities. We use the Snake and Ant robots in Gather. Details for all robotic agents are provided in Appendix B.

## 5.2 LEARNING FROM SCRATCH AND TIME-COMMITMENT

In this section, we study the benefit of using our HiPPO algorithm instead of standard PPO on a flat policy (Schulman et al., 2017). The results, reported in Figure 3, demonstrate that training from scratch with HiPPO leads to faster learning and better performance than flat PPO. Furthermore, we show that the benefit of HiPPO does not just come from having temporally correlated exploration: PPO with action repeat converges at a lower performance than our method. HiPPO leverages the time-commitment more efficiently, as suggested by the poor performance of the ablation where we set $p = 1$, when the manager takes an action every environment step as well. Finally, Figure 4 shows the effectiveness of using the presented skill-dependent baseline.

## 5.3 COMPARISON TO OTHER METHODS

We compare HiPPO to current state-of-the-art hierarchical methods. First, we evaluate HIRO (Nachum et al., 2018), an off-policy RL method based on training a goal-reaching lower level policy. Fig. 5 shows that HIRO achieves poor performance on our tasks. As further detailed in Appendix D, this algorithm is sensitive to access to ground-truth information, like the exact $(x, y)$ position of the robot in Gather. In contrast, our method is able to perform well directly from the raw sensory inputs described in Section 5.1. We evaluate Option-Critic (Bacon et al., 2017), a variant of the options framework (Sutton et al., 1999) that can be used for continuous action-spaces. It fails to learn, and we hypothesize that their algorithm provides less time-correlated exploration and learns

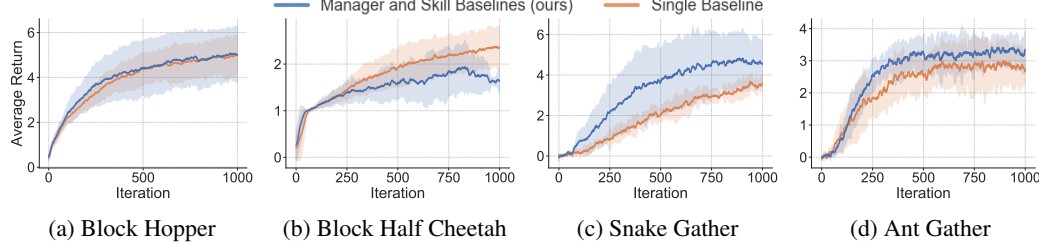

(a) Block Hopper    (b) Block Half Cheetah    (c) Snake Gather    (d) Ant Gather

Figure 4: Using a skill-conditioned baseline, as defined in Section 4.2, generally improves performance of HiPPO when learning from scratch.

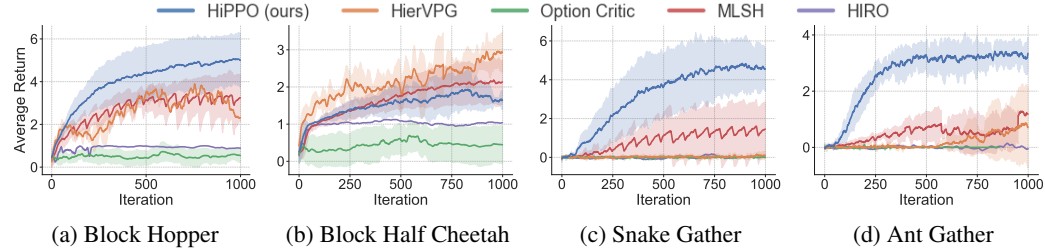

(a) Block Hopper    (b) Block Half Cheetah    (c) Snake Gather    (d) Ant Gather

Figure 5: Comparison of HiPPO and HierVPG to prior hierarchical methods on learning from scratch.

less diverse skills. We also compare against MLSH (Frans et al., 2018), which repeatedly samples new environment configurations to learn primitive skills. We take these hyperparameters from their Ant Twowalk experiment: resetting the environment configuration every 60 iterations, a warmup period of 20 during which only the manager is trained, and a joint training period of 40 during which both manager and skills are trained. Our results show that such a training scheme does not provide any benefits. Finally, we provide a comparison to a direct application of our Hierarchical Vanilla Policy Gradient (HierVPG) algorithm, and we see that the algorithm is unstable without PPO's trust-region-like technique.

## 5.4 ROBUSTNESS TO DYNAMICS PERTURBATIONS

We investigate the robustness of HiPPO to changes in the dynamics of the environment. We perform several modifications to the base Snake Gather and Ant Gather environments. One at a time, we change the body mass, dampening of the joints, body inertia, and friction characteristics of both robots. The results, presented in Table 1, show that HiPPO with randomized period $\texttt{Categorical}([T_{\min}, T_{\max}])$ is able to better handle these dynamics changes. In terms of the drop in policy performance between the training environment and test environment, it outperforms HiPPO with fixed period on 6 out of 8 related tasks. These results suggest that the randomized period exposes the policy to a wide range of scenarios, which makes it easier to adapt when the environment changes.

| Gather | Algorithm | Initial | Mass | Dampening | Inertia | Friction |
|--------|-----------|---------|------|-----------|---------|----------|
| Snake  | Flat PPO | 2.72 | 3.16 (+16%) | 2.75 (+1%) | 2.11 (-22%) | 2.75 (+1%) |
|        | HiPPO, $p = 10$ | 4.38 | 3.28 (-25%) | 3.27 (-25%) | 3.03 (-31%) | 3.27 (-25%) |
|        | HiPPO random $p$ | 5.11 | **4.09** (-20%) | **4.03** (-21%) | **3.21** (-37%) | **4.03** (-21%) |
| Ant    | Flat PPO | 2.25 | 2.53 (+12%) | 2.13 (-5%) | 2.36 (+5%) | 1.96 (-13%) |
|        | HiPPO, $p = 10$ | 3.84 | 3.31 (-14%) | **3.37** (-12%) | 2.88 (-25%) | **3.07** (-20%) |
|        | HiPPO random $p$ | 3.22 | **3.37** (+5%) | 2.57 (-20%) | **3.36** (+4%) | 2.84 (-12%) |

Table 1: Zero-shot transfer performance. The final return in the initial environment is shown, as well as the average return over 25 rollouts in each new modified environment.

## 5.5 ADAPTATION OF PRE-TRAINED SKILLS

For the Block task, we use DIAYN (Eysenbach et al., 2019) to train 6 differentiated subpolicies in an environment without any walls. Here, we see if these diverse skills can improve performance on a downstream task that's out of the training distribution. For Gather, we take 6 pretrained

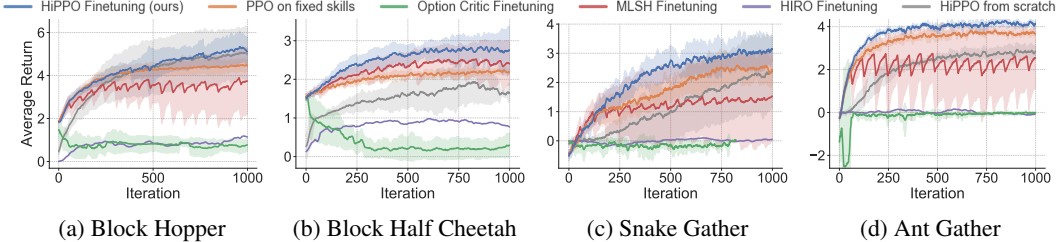

Figure 6: Benefit of adapting some given skills when the preferences of the environment are different from those of the environment where the skills were originally trained. Adapting skills with HiPPO has better learning performance than leaving the skills fixed or learning from scratch.

subpolicies encoded by a Stochastic Neural Network (Tang & Salakhutdinov, 2013) that was trained in a diversity-promoting environment (Florensa et al., 2017a). We fine-tune them with HiPPO on the Gather environment, but with an extra penalty on the velocity of the Center of Mass. This can be understood as a preference for cautious behavior. This requires adjustment of the sub-policies, which were trained with a proxy reward encouraging them to move as far as possible (and hence quickly). Fig. 6 shows that using HiPPO to simultaneously train a manager and fine-tune the skills achieves higher final performance than fixing the sub-policies and only training a manager with PPO. The two initially learn at the same rate, but HiPPO's ability to adjust to the new dynamics allows it to reach a higher final performance. Fig. 6 also shows that HiPPO can fine-tune the same given skills better than Option-Critic (Bacon et al., 2017), MLSH (Frans et al., 2018), and HIRO (Nachum et al., 2018).

## 5.6 SKILL DIVERSITY ASSUMPTION

In Lemma 1, we derived a more efficient and numerically stable gradient by assuming that the sub-policies are diverse. In this section, we empirically test the validity of our assumption and the quality of our approximation. We run the HiPPO algorithm on Ant Gather and Snake Gather both from scratch and with given pretrained skills, as done in the previous section. In Table 2, we report the average maximum probability under other sub-policies, corresponding to $\epsilon$ from the assumption. In all settings, this is on the order of magnitude of 0.1. Therefore, under the $p \approx 10$ that we use in our experiments, the term we neglect has a factor $\epsilon^{p-1} = 10^{-10}$. It is not surprising then that the average cosine similarity between the full gradient and our approximation is almost 1, as reported in Table 2.

| Gather | Algorithm | Cosine Sim. | $\max_{z' \neq z_{kp}} \pi_{\theta_l}(a_t \mid s_t, z')$ | $\pi_{\theta_l}(a_t \mid s_t, z_{kp})$ |
|---|---|---|---|---|
| Snake | HiPPO on given skills | $0.98 \pm 0.01$ | $0.09 \pm 0.04$ | $0.44 \pm 0.03$ |
|  | HiPPO on random skills | $0.97 \pm 0.03$ | $0.12 \pm 0.03$ | $0.32 \pm 0.04$ |
| Ant | HiPPO on given skills | $0.96 \pm 0.04$ | $0.11 \pm 0.05$ | $0.40 \pm 0.08$ |
|  | HiPPO on random skills | $0.94 \pm 0.03$ | $0.13 \pm 0.05$ | $0.31 \pm 0.09$ |

Table 2: Empirical evaluation of Lemma 1. In the middle and right columns, we evaluate the quality of our assumption by computing the largest probability of a certain action under other skills ($\epsilon$), and the action probability under the actual latent. We also report the cosine similarity between our approximate gradient and the exact gradient from Eq. 3. The mean and standard deviation of these values are computed over the full batch collected at iteration 10.

## 6 CONCLUSIONS AND FUTURE WORK

In this paper, we examined how to effectively adapt temporal hierarchies. We began by deriving a hierarchical policy gradient and its approximation. We then proposed a new method, HiPPO, that can stably train multiple layers of a hierarchy jointly. The adaptation experiments suggest that we can optimize pretrained skills for downstream environments, and learn emergent skills without any unsupervised pre-training. We also demonstrate that HiPPO with randomized period can learn from scratch on sparse-reward and long time horizon tasks, while outperforming non-hierarchical methods on zero-shot transfer.

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

## A  Hyperparameters and Architectures

The Block environments used a horizon of 1000 and a batch size of 50,000, while Gather used a batch size of 100,000. Ant Gather has a horizon of 5000, while Snake Gather has a horizon of 8000 due to its larger size. For all experiments, both PPO and HiPPO used learning rate $3 \times 10^{-3}$, clipping parameter $\epsilon = 0.1$, 10 gradient updates per iteration, and discount $\gamma = 0.999$. The learning rate, clipping parameter, and number of gradient updates come from the OpenAI Baselines implementation.

HiPPO used $n = 6$ sub-policies. HiPPO uses a manager network with 2 hidden layers of 32 units, and a skill network with 2 hidden layers of 64 units. In order to have roughly the same number of parameters for each algorithm, flat PPO uses a network with 2 hidden layers with 256 and 64 units respectively. For HiPPO with randomized period, we resample $p \sim \text{Uniform}\{5, 15\}$ every time the manager network outputs a latent, and provide the number of timesteps until the next latent selection as an input into both the manager and skill networks. The single baselines and skill-dependent baselines used a MLP with 2 hidden layers of 32 units to fit the value function. The skill-dependent baseline receives, in addition to the full observation, the active latent code and the time remaining until the next skill sampling. All runs used five random seeds.

## B  Robot Agent Description

Hopper is a 3-link robot with a 14-dimensional observation space and a 3-dimensional action space. Half-Cheetah has a 20-dimensional observation space and a 6-dimensional action space. We evaluate both of these agents on a sparse block hopping task. In addition to observing their own joint angles and positions, they observe the height and length of the next wall, the x-position of the next wall, and the distance to the wall from the agent. We also provide the same wall observations for the previous wall, which the agent can still interact with.

Snake is a 5-link robot with a 17-dimensional observation space and a 4-dimensional action space. Ant is a quadrupedal robot with a 27-dimensional observation space and a 8-dimensional action space. Both Ant and Snake can move and rotate in all directions, and Ant faces the added challenge of avoiding falling over irrecoverably. In the Gather environment, agents also receive 2 sets of 10-dimensional lidar observations, whcih correspond to separate apple and bomb observations. The observation displays the distance to the nearest apple or bomb in each $36°$ bin, respectively. All environments are simulated with the physics engine MuJoCo (Todorov et al., 2012).

## C  Proofs

**Lemma 1.** If the skills are sufficiently differentiated, then the latent variable can be treated as part of the observation to compute the gradient of the trajectory probability. Concretely, if $\pi_{\theta_h}(z|s)$ and $\pi_{\theta_l}(a|s, z)$ are Lipschitz in their parameters, and $0 < \pi_{\theta_l}(a_t|s_t, z_j) < \epsilon \ \forall j \neq kp$, then

$$\nabla_\theta \log P(\tau) = \sum_{k=0}^{H/p} \nabla_\theta \log \pi_{\theta_h}(z_{kp}|s_{kp}) + \sum_{t=1}^{p} \nabla_\theta \log \pi_{\theta_l}(a_t|s_t, z_{kp}) + \mathcal{O}(nH\epsilon^{p-1}) \quad (5)$$

*Proof.* From the point of view of the MDP, a trajectory is a sequence $\tau = (s_0, a_0, s_1, a_1, \dots, a_{H-1}, s_H)$. Let's assume we use the hierarchical policy introduced above, with a higher-level policy modeled as a parameterized discrete distribution with $n$ possible outcomes $\pi_{\theta_h}(z|s) = Categorical_{\theta_h}(n)$. We can expand $P(\tau)$ into the product of policy and environment dynamics terms, with $z_j$ denoting the $j$th possible value out of the $n$ choices,

$$P(\tau) = \left( \prod_{k=0}^{H/p} \left[ \sum_{j=1}^{n} \pi_{\theta_h}(z_j|s_{kp}) \prod_{t=kp}^{(k+1)p-1} \pi_{\theta_l}(a_t|s_t, z_j) \right] \right) \left[ P(s_0) \prod_{t=1}^{H} P(s_{t+1}|s_t, a_t) \right]$$

Taking the gradient of $\log P(\tau)$ with respect to the policy parameters $\theta = [\theta_h, \theta_l]$, the dynamics terms disappear, leaving:

$$\nabla_\theta \log P(\tau) = \sum_{k=0}^{H/p} \nabla_\theta \log \Big( \sum_{j=1}^{n} \pi_{\theta_l}(z_j|s_{kp}) \prod_{t=kp}^{(k+1)p-1} \pi_{s,\theta}(a_t|s_t, z_j) \Big)$$

$$= \sum_{k=0}^{H/p} \frac{1}{\sum_{j=1}^{n} \pi_{\theta_h}(z_j|s_{kp}) \prod_{t=kp}^{(k+1)p-1} \pi_{\theta_l}(a_t|s_t, z_j)} \sum_{j=1}^{n} \nabla_\theta \Big( \pi_{\theta_h}(z_j|s_{kp}) \prod_{t=kp}^{(k+1)p-1} \pi_{\theta_l}(a_t|s_t, z_j) \Big)$$

The sum over possible values of $z$ prevents the logarithm from splitting the product over the $p$-step sub-trajectories. This term is problematic, as this product quickly approaches 0 as $p$ increases, and suffers from considerable numerical instabilities. Instead, we want to approximate this sum of products by a single one of the terms, which can then be decomposed into a sum of logs. For this we study each of the terms in the sum: the gradient of a sub-trajectory probability under a specific latent $\nabla_\theta \Big( \pi_{\theta_h}(z_j|s_{kp}) \prod_{t=kp}^{(k+1)p-1} \pi_{\theta_l}(a_t|s_t, z_j) \Big)$. Now we can use the assumption that the skills are easy to distinguish, $0 < \pi_{\theta_l}(a_t|s_t, z_j) < \epsilon \ \forall j \neq kp$. Therefore, the probability of the sub-trajectory under a latent different than the one that was originally sampled $z_j \neq z_{kp}$, is upper bounded by $\epsilon^p$. Taking the gradient, applying the product rule, and the Lipschitz continuity of the policies, we obtain that for all $z_j \neq z_{kp}$,

$$\nabla_\theta \Big( \pi_{\theta_h}(z_j|s_{kp}) \prod_{t=kp}^{(k+1)p-1} \pi_{\theta_l}(a_t|s_t, z_j) \Big) = \nabla_\theta \pi_{\theta_h}(z_j|s_{kp}) \prod_{t=kp}^{(k+1)p-1} \pi_{\theta_l}(a_t|s_t, z_j) +$$

$$\sum_{t=kp}^{(k+1)p-1} \pi_{\theta_h}(z_j|s_{kp}) \big( \nabla_\theta \pi_{\theta_l}(a_t|s_t, z_j) \big) \prod_{\substack{t=kp \\ t' \neq t}}^{(k+1)p-1} \pi_{\theta_l}(a_{t'}|s_{t'}, z_j)$$

$$= \mathcal{O}(p\epsilon^{p-1})$$

Thus, we can across the board replace the summation over latents by the single term corresponding to the latent that was sampled at that time.

$$\nabla_\theta \log P(\tau) = \sum_{k=0}^{H/p} \frac{1}{\pi_{\theta_h}(z_{kp}|s_{kp}) \prod_{t=kp}^{(k+1)p-1} \pi_{\theta_l}(a_t|s_t, z_{kp})} \nabla_\theta \Big( P(z_{kp}|s_{kp}) \prod_{t=kp}^{(k+1)p-1} \pi_{\theta_l}(a_t|s_t, z_{kp}) \Big) + \frac{nH}{p} \mathcal{O}(p\epsilon^{p-1})$$

$$= \sum_{k=0}^{H/p} \nabla_\theta \log \Big( \pi_{\theta_h}(z_{kp}|s_{kp}) \prod_{t=kp}^{(k+1)p-1} \pi_{\theta_l}(a_t|s_t, z_{kp}) \Big) + \mathcal{O}(nH\epsilon^{p-1})$$

$$= \mathbb{E}_\tau \Big[ \Big( \sum_{k=0}^{H/p} \nabla_\theta \log \pi_{\theta_h}(z_{kp}|s_{kp}) + \sum_{t=1}^{H} \nabla_\theta \log \pi_{\theta_l}(a_t|s_t, z_{kp}) \Big) \Big] + \mathcal{O}(nH\epsilon^{p-1})$$

Interestingly, this is exactly $\nabla_\theta P(s_0, z_0, a_0, s_1, \dots)$. In other words, it's the gradient of the probability of that trajectory, where the trajectory now includes the variables $z$ as if they were observed.

$\square$

**Lemma 2.** For any functions $b_h : \mathcal{S} \to \mathbb{R}$ and $b_l : \mathcal{S} \times \mathcal{Z} \to \mathbb{R}$ we have:

$$\mathbb{E}_\tau [\sum_{k=0}^{H/p} \nabla_\theta \log P(z_{kp}|s_{kp}) b(s_{kp})] = 0$$

$$\mathbb{E}_\tau [\sum_{t=0}^{H} \nabla_\theta \log \pi_{\theta_l}(a_t|s_t, z_{kp}) b(s_t, z_{kp})] = 0$$

*Proof.* We can use the tower property as well as the fact that the interior expression only depends on $s_{kp}$ and $z_{kp}$:

$$\mathbb{E}_\tau[\sum_{k=0}^{H/p} \nabla_\theta \log P(z_{kp}|s_{kp})b(s_{kp})] = \sum_{k=0}^{H/p} \mathbb{E}_{s_{kp},z_{kp}}[\mathbb{E}_{\tau\backslash s_{kp},z_{kp}}[\nabla_\theta \log P(z_{kp}|s_{kp})b(s_{kp})]]$$

$$= \sum_{k=0}^{H/p} \mathbb{E}_{s_{kp},z_{kp}}[\nabla_\theta \log P(z_{kp}|s_{kp})b(s_{kp})]$$

Then, we can write out the definition of the expectation and undo the gradient-log trick to prove that the baseline is unbiased.

$$\mathbb{E}_\tau[\sum_{k=0}^{H/p} \nabla_\theta \log \pi_{\theta_h}(z_{kp}|s_{kp})b(s_{kp})] = \sum_{k=0}^{H/p} \int_{(s_{kp},z_{kp})} P(s_{kp},z_{kp})\nabla_\theta \log \pi_{\theta_h}(z_{kp}|s_{kp})b(s_{kp})dz_{kp}ds_{kp}$$

$$= \sum_{k=0}^{H/p} \int_{s_{kp}} P(s_{kp})b(s_{kp}) \int_{z_{kp}} \pi_{\theta_h}(z_{kp}|s_{kp})\nabla_\theta \log \pi_{\theta_h}(z_{kp}|s_{kp})dz_{kp}ds_{kp}$$

$$= \sum_{k=0}^{H/p} \int_{s_{kp}} P(s_{kp})b(s_{kp}) \int_{z_{kp}} \pi_{\theta_h}(z_{kp}|s_{kp})\frac{1}{\pi_{\theta_h}(z_{kp}|s_{kp})}\nabla_\theta \pi_{\theta_h}(z_{kp}|s_{kp})dz_{kp}ds_{kp}$$

$$= \sum_{k=0}^{H/p} \int_{s_{kp}} P(s_{kp})b(s_{kp})\nabla_\theta \int_{z_{kp}} \pi_{\theta_h}(z_{kp}|s_{kp})dz_{kp}ds_{kp}$$

$$= \sum_{k=0}^{H/p} \int_{s_{kp}} P(s_{kp})b(s_{kp})\nabla_\theta 1 ds_{kp}$$

$$= 0$$

$\square$

Subtracting a state- and subpolicy- dependent baseline from the second term is also unbiased, i.e.

$$\mathbb{E}_\tau[\sum_{t=0}^{H} \nabla_\theta \log \pi_{s,\theta}(a_t|s_t, z_{kp})b(s_t, z_{kp})] = 0$$

We'll follow the same strategy to prove the second equality: apply the tower property, express the expectation as an integral, and undo the gradient-log trick.

$$\mathbb{E}_\tau[\sum_{t=0}^{H} \nabla_\theta \log \pi_{\theta_l}(a_t|s_t, z_{kp})b(s_t, z_{kp})]$$

$$= \sum_{t=0}^{H} \mathbb{E}_{s_t,a_t,z_{kp}}[\mathbb{E}_{\tau\backslash s_t,a_t,z_{kp}}[\nabla_\theta \log \pi_{\theta_m}(a_t|s_t, z_{kp})b(s_t, z_{kp})]]$$

$$= \sum_{t=0}^{H} \mathbb{E}_{s_t,a_t,z_{kp}}[\nabla_\theta \log \pi_{\theta_l}(a_t|s_t, z_{kp})b(s_{kp}, z_{kp})]$$

$$= \sum_{t=0}^{H} \int_{(s_t,z_{kp})} P(s_t, z_{kp})b(s_t, z_{kp}) \int_{a_t} \pi_{\theta_l}(a_t|s_t, z_{kp})\nabla_\theta \log \pi_{\theta_l}(a_t|s_t, z_{kp})da_t dz_{kp}ds_t$$

$$= \sum_{t=0}^{H} \int_{(s_t,z_{kp})} P(s_t, z_{kp})b(s_t, z_{kp})\nabla_\theta 1 dz_{kp}ds_t$$

$$= 0$$

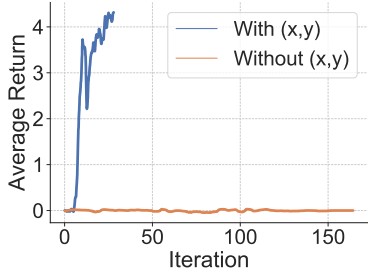

Figure 7: HIRO performance on Ant Gather with and without access to the ground truth $(x, y)$, which it needs to communicate useful goals.

## D   HIRO SENSITIVITY TO OBSERVATION-SPACE

In this section we provide a more detailed explanation of why HIRO (Nachum et al., 2018) performs poorly under our environments. As explained in our related work section, HIRO belongs to the general category of algorithms that train goal-reaching policies as lower levels of the hierarchy (Vezhnevets et al., 2017; Levy et al., 2017). These methods rely on having a goal-space that is meaningful for the task at hand. For example, in navigation tasks they require having access to the $(x, y)$ position of the agent such that deltas in that space can be given as meaningful goals to move in the environment. Unfortunately, in many cases the only readily available information (if there's no GPS signal or other positioning system installed) are raw sensory inputs, like cameras or the LIDAR sensors we mimic in our environments. In such cases, our method still performs well because it doesn't rely on the goal-reaching extra supervision that is leveraged (and detrimental in this case) in HIRO and similar methods. In Figure 7, we show that knowing the ground truth location is critical for its success. We have reproduced the HIRO results in Fig. 7 using the published codebase, so we are convinced that our results showcase a failure mode of HIRO.

## E   HYPERPARAMETER SENSITIVITY PLOTS

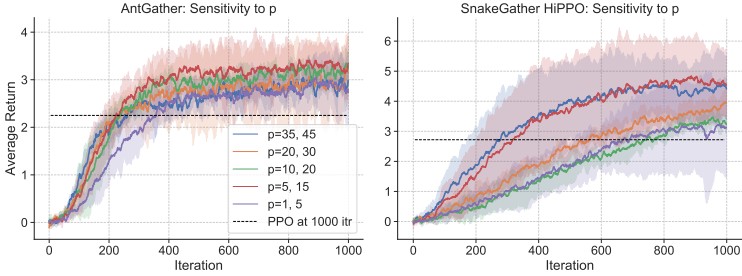

Figure 8: Sensitivity of HiPPO to variation in the time-commitment.

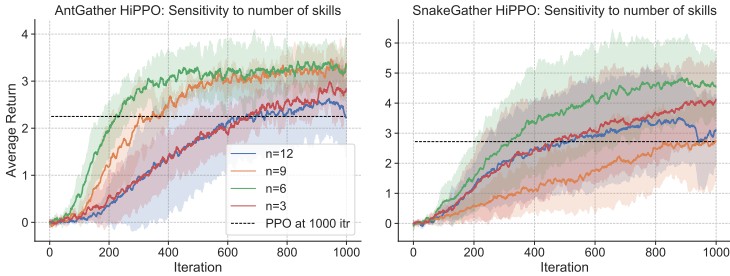

Figure 9: Sensitivity of HiPPO to variation in the number of skills.

