# OpenReview forum: "Sub-policy Adaptation for Hierarchical Reinforcement Learning"
_ICLR.cc/2020/Conference — Accept (Poster)_

### Official Review · AnonReviewer1 · 2019-10-19
**Official Blind Review #564**

**Rating:** 8

**Review:**

The main problem they try to tackle is to train agents for unseen tasks and environmental changes. They show that their method has a better performance and is more robust against sensor errors and physical parameter alterations.

The authors clearly position their work in the HRL paradigm and explain current limitations/challenges within that field. Alike other HRL agents, their method has two types of policies (manager and subpolicies), but different from other works they do not keep parameters fixed in post training for new tasks. In addition to the parameters, they do not fix the time length.

The paper is very well written, clearly stated the contributions.

Remarks:
- Fig 2 has no caption. How are the colors of balls obtained, since they only explain how the sensors (lidar) measure distances to balls (bombs/apples).
- Fig 4/5a, some agents (blue) seems to have undesired behaviour (until half of the iterations). This behaviour is not described anywhere.
- The URL of the website with code and videos does not have any code.

Questions to the authors:
- Closest work is Frans et al. (2018). The experiments do not show Frans et al. as a benchmark method. Why?
- HiPPO shows to have a higher robustness. Why are the results of different methods (p=10/random) in Table 1 for different environments (Snake/Ant) different?




**Experience Assessment:**

I do not know much about this area.

**Review Assessment: Checking Correctness Of Derivations And Theory:**

I did not assess the derivations or theory.

**Review Assessment: Checking Correctness Of Experiments:**

I did not assess the experiments.

**Review Assessment: Thoroughness In Paper Reading:**

I made a quick assessment of this paper.

---

> ### Author Response · Authors · 2019-11-14
> **Additional details and new benchmark (MLSH)**
>
> We thank the reviewer for their positive review, and for providing remarks and questions that have helped us further improve our work. Here is a summary of the modifications:
> - We have updated the caption for Fig. 2. There are separate lidar signatures for bombs and apples, so agents can indeed distinguish between the two. This is now better explained in the text, and in Appendix B.
> - We have updated Fig. 4 and 6 [previously Fig. 5]. The undesired behavior was due to uncomplete runs of that agent. Please let us know if any confusion remains.
> - The website contains a link (“See implementation here”) to an anonymized Github repo that contains code for our experiments.
> - We have added comparisons to MLSH in Fig. 4 and Fig. 6. Our results show that the MLSH training scheme does not help it learn better from scratch or when fine-tuning pre-trained skills in these tasks.
> - HiPPO with p=10 learns slightly better on Ant than HiPPO with random period. However, Table 1 shows that the percent change in performance is better for randomized period, leading HiPPO to outperform its fixed p=10 counterpart in 6/8 overall scenarios.
>
> We hope we have addressed all of your concerns, and if any remain please let us know.

---

### Official Review · AnonReviewer2 · 2019-10-22
**Official Blind Review #2**

**Rating:** 3

**Review:**

This paper is under the topic of hierarchical reinforcement learning. The motivation of this paper is "most methods still decouple the lower-level skill acquisition process and the training of a higher level that controls the skills in a new task." The paper proposes a method to learn higher-level skill selection and lower-level skill improvement jointly.

What I like in this paper:
    1. The paper, in general, is well-written so that I can understand it well.
    2. Experiments are question-driven and provide interesting results.
    3. Theories are closely related to the algorithm.

Key reasons for my rejection:
    1. My biggest concern is the motivation of this paper.
    The joint learning of higher-level policy and lower-level skill discovery is not rare in modern literature. Some works are even cited in this paper, for example, option-critic, feudal network, etc. These methods fix their skills in the new task, not because they are inherently not able to do so, but because they want to demonstrate that the learned skills can be reused in new tasks, even if there is no further adaptation. I agree with the author that the agent needs to adapt its skills when faced with new tasks. But I don't think most works are limited in this aspect, as claimed by the paper in the abstract.

    2. I think the author didn't justify his key design choices well.
    This paper is under the research area of "hierarchical reinforcement learning." However, just like temporal abstraction, the HRL is a general idea instead of an existing problem formulation or a particular algorithm. It seems that the author is not aware of this point as the paper claims a particular way of achieving HRL is the HRL itself (in section 4.1 "In the context of HRL, a hierarchical policy with a manager πθh(zt|st) selects every p time-steps one of n sub-policies to execute."). I would like to see the paper takes the responsibility to justify the reason it follows this particular way. There are two more key decisions the paper proposed but not fully justified and analyzed.
        1. Why is random length a valid choice? The paper doesn't tell readers the consequence of this design choice. For example, what about the optimality of the solution? Since bounding the random length needs prior knowledge, how difficult is it to come up with the prior knowledge. Is the algorithm sensitive to prior knowledge?
        2. Why is it fine to assume "for each action, there is just one sub-policy that gives it high probability"? What would be the consequence of this assumption? Well, the extreme case is each action is only being chosen by one sub-policy. Therefore, executing the sub-policy becomes executing a repeated sequence of the same action. Obviously, this is a kind of temporal abstraction but is a very limited one.

Other small issues:
Section 2: "... maximize the discounted expected reward ..." should be "... maximize the discounted expected return ...".
Section 2: the horizon T in the definition in \eta should be H.
Section 4: the advantage function is not defined.

**Experience Assessment:**

I have read many papers in this area.

**Review Assessment: Checking Correctness Of Derivations And Theory:**

I assessed the sensibility of the derivations and theory.

**Review Assessment: Checking Correctness Of Experiments:**

I assessed the sensibility of the experiments.

**Review Assessment: Thoroughness In Paper Reading:**

I read the paper at least twice and used my best judgement in assessing the paper.

---

> ### Author Response · Authors · 2019-11-14
> **Clarifying motivation, impact, and design choices.**
>
> We thank the reviewer #2 for their detailed comments - they have helped improve our exposition of the motivations, design choices, and further impact of our work.
>
> *Response to concern 1*:
> We agree with Reviewer #2 that there exists some recent work in HRL that allows for jointly training both levels of the hierarchy. Nevertheless, this is still not the norm, and many recent HRL works have a separate procedure to train skills and don’t show any fine-tuning: DADS (Sharma et al. 2019), DIAYN (Eysenbach et al. 2018), SNN4HRL (Florensa et al. 2017), etc. Our method can enhance any such two-step method, as shown in our Fig. 6. [originally Fig.5], introducing a principled way to adapt given skills learned in the first step. Any work using these techniques will benefit from our study.
>
> Furthermore, the end-to-end HRL methods cited by the reviewer, and all prior work known to us, suffer from either collapsing skills (e.g. Option-Critic, Bacon et al. 2017, and other option-based approaches), or from setting different reward functions for the lower level and higher-level parts of the hierarchy (HIRO, Nachum et al. 2018; FuN, Vezhnevetz et al. 2017, etc.). These approaches might hinder the learning performance in tasks that cannot be described as goal-reaching.  We can see the limitations on learning from scratch of other “end-to-end” methods in Fig. 5 [originally Fig. 6 in the Appendix], where we show that both Option-Critic and HIRO (which has been shown to outperform FuN) greatly struggle in the more challenging tasks where our HiPPO algorithm shines. We have moved the figure to the main text, and have updated it with an additional comparison with MLSH (Frans et al. 2018) as Reviewer #564 suggested.
>
> In Fig. 6, we compare the fine-tuning performance of HiPPO to prior hierarchical approaches, and we again find that HiPPO has better sample efficiency and asymptotic performance than Option-Critic, MLSH, and HIRO. We observe quick skill collapse for Option-Critic, and HIRO’s goal-reaching formulation fails to learn well on all four tasks.
>
> *Response to concern 2*:
> We agree with the reviewer that HRL encompasses a wide variety of approaches. We have clarified that our work focuses on improving the training of temporal hierarchies for problems that have been shown to benefit from it, like the long horizon locomotion+navigation tasks we tackle in our work. Let us know if we should make any more clarifications.
>
> In terms of the two design choices made in our paper:
> - We propose the random length to avoid learning skills that overfit to a fixed time-switch, which makes the skill more brittle when deployed in an environment with different dynamics. The benefit of this design choice in zero-shot transfer is empirically demonstrated in our Table 1. We have also included hyperparameter sensitivity plots in the Appendix, which show that our method performs well for a large range of sensible time commitments - hence requiring little prior knowledge. Any temporal hierarchy can theoretically be suboptimal; however, our experiments show that despite the potential sub-optimality, HiPPO learns better policies than other methods without the random time-commitment.
> - The assumption of our lemma is actually the following: conditioned on the state, the policy action is given high probability by only one skill. The skills don’t partition the action-space into constant sets for all states; rather, different skills fully utilize the same action space to compose sequences of actions into very different primitives. Our assumption is actually quite mild, as we show empirically in Table 2. Even HiPPO trained on random skills satisfies it!
>
> We hope we have addressed all of your concerns, and if any remain please let us know.

---

> > ### Comment · AnonReviewer2 · 2019-11-15
> > **Response to the author's clarification**
> >
> > 1. The authors wrote in the response that "These approaches (option-critic, FuN, HIRO) might hinder the learning performance in tasks that cannot be described as goal-reaching." I guess the "goal-reaching" here means reaching a particular state or group of states. I agree with the authors that there are problems that are not goal-reaching. And I agree that FuN and HIRO are limited to goal-reaching problems (they are in fact, even more restricted as they assume the euclidian distance between states can be defined and would be good to use to define pseudo rewards). However, I think option-critic method and its related works are not limited to the goal-reaching case.
> >
> > 2. In the response, the authors pointed out the option-critic suffers from collapsing skill problem and thus performs worse than the proposed HiPPO, as shown in figure 5, 6.  However, I am doubtful about this point. 1) From both theoretical and empirical parts of the paper, I don't see any reason the proposed HiPPO is immune to the collapsing skill problem. In fact, I would guess it also suffers from the same problem as there is no mechanism in the algorithm preventing this from happening. 2) The collapsing skills problem doesn't necessarily lead to bad performance. With one skill, the option-critic degenerates to the actor-critic, still a decent RL algorithm.
> >
> > 3. I notice that the authors made a change in section 4.1: "In a temporal hierarchy, a hierarchical policy with a manager πθh (zt|st) selects every p time-steps one of n sub-policies to execute." I appreciate the authors to make this change, but unfortunately, this is not what I suggested. That "a hierarchical policy with a manager πθh (zt|st) selects every p time-steps one of n sub-policies to execute" is a specific temporal hierarchy. It is not the only possible temporal hierarchy even for "long horizon locomotion+navigation tasks" tackled in this paper. And this particular temporal hierarchy is less general than some other temporal hierarchies like the option framework. So why should we follow this particular way? For example, is that because that makes the algorithm design easier? What would be the consequence of this simplification? It would be great for me and other readers to understand if the authors could provide reasons for this key design choice, or refers to other works which make the same design choice and give reasonable justifications.

---

> > > ### Author Response · Authors · 2019-11-15
> > > **Further clarification of time-commitment**
> > >
> > > 1. We are glad the reviewer agrees with our statement that many of the recent end-to-end hierarchical methods (FuN, HIRO, etc.) are limited to goal-reaching problems. We agree with the reviewer that Option-Critic does not fall into this category, and we hope it's clear in our updated paper.
> > >
> > > 2. HiPPO actually does have mechanisms to prevent skill collapse to atomic actions. First, with the random time-commitment, HiPPO does NOT train a termination function, so each skill must be coherent and useful for about 15 steps, which makes learning long-horizon strategies easier. HiPPO also learns at two timescales: a fine-grained timescale for training the skills, and a coarser timescale for training the manager. In contrast, Option-Critic learns both the q-function and the termination function at the fine-grained timescale, which means that it might not be easier to learn high-level decision-making. Another novel factor that helps HiPPO optimize at two different timescales is our introduction of different baselines for the manager and the skills. Furthermore, HiPPO’s use of PPO likelihood clipping prevents drastic changes in policy behavior.
> > >
> > > We also agree that in the worst-case scenario Option-Critic becomes simple Actor-Critic. However, Actor-Critic (and flat policies in general) lack the benefits that a temporal hierarchy can confer in terms of effective and temporally correlated exploration. Poor results for PPO and Option-Critic on our environments demonstrate this failure case. We agree that skill-collapse is not always fatal: we have observed Option-Critic learning quite well on simpler environments, such as Point-Mass Navigation, Cartpole Balance, and Cartpole Swingup.
> > >
> > > 3. We thank the reviewer for the additional details about their concern, and yes, we fully agree that the temporal hierarchy that we study is a specific one, not the most general case. We have corrected the wording in the paper to make this explicit.
> > > Nevertheless, we argue that a fixed time-commitment is quite common in the literature (HIRO, FuN, SNN4HRL, DADS, etc.). The impact of this design decision depends on the environment we use it in. In tasks where rapid, short-timescale control is important, our fixed or random time-commitment might be noticeably suboptimal. However, for the long-horizon problems we tackle, our adaptation strategy confers several benefits, as detailed in the second topic above.
> > >
> > > Empirically, we see that our time-commitment strategy does better than allowing arbitrary switching between skills. Results on all four environments show that HiPPO outperforms Option-Critic in both learning from scratch and fine-tuning skills. Again, hyperparameter sensitivity plots in the Appendix show that HiPPO achieves high performance for a wide range of time-commitments. We do have a relevant ablation in Fig. 3: HiPPO p=1, which means that the manager chooses an active skill at every timestep. On the Block environments, where high-level decision making is not as important, HiPPO p=1 does on par with our algorithm. In the Gather environment, where it’s difficult to choose the optimal route for collecting the most apples over a horizon of 5000 - 8000, HiPPO p=1 performs worse. Finally, as discussed in the paper, the fact that the time-commitment strategy is not learned simplifies the gradient (since there’s no termination function) and acts as a regularizer for the robustness of the final skills.
> > >
> > > We believe our proofs and strong empirical findings are of high interest to the research community. We find the reviewer’s responses extremely helpful, and hope that we have addressed all of the remaining concerns. Please let us know otherwise.

---

### Decision · Program_Chairs · 2019-12-19

**Decision:**

Accept (Poster)

**Comment:**

This paper considers hierarchical reinforcement learning, and specifically the case where the learning and use of lower-level skills should not be decoupled. To this end the paper proposes Hierarchical Proximal Policy Optimization (HiPPO) to jointly learn the different layers of the hierarchy. This is compared against other hierarchical RL schemes on several Mujoco domains.

The reviewers raised three main issues with this paper. The first concerns an excluded baseline, which was included in the rebuttal. The other issues involve the motivation for the paper (in that there exist other methods that try and learn different levels of hierarchy together) and justification for some design choices. These were addressed to some extent in the rebuttal, but I believe this to still be an interesting contribution to the literature, and should be accepted.